# Drinking Warm Water Promotes Performance by Regulating Ruminal Microbial Composition and Serum Metabolites in Yak Calves

**DOI:** 10.3390/microorganisms11082092

**Published:** 2023-08-16

**Authors:** Tianxu Liu, Qianqian Wang, Chenxi Gao, Shenfei Long, Tengfei He, Zhenlong Wu, Zhaohui Chen

**Affiliations:** State Key Laboratory of Animal Nutrition and Feeding, College of Animal Science and Technology, China Agricultural University, Beijing 100193, China; liutianx@cau.edu.cn (T.L.); wqq@cau.edu.cn (Q.W.); 17606944398@163.com (C.G.); longshenfei@cau.edu.cn (S.L.); hetengfei@cau.edu.cn (T.H.); wuzhenlong@cau.edu.cn (Z.W.)

**Keywords:** yak calves, warm water, growth performance, ruminal fermentation characteristics, ruminal microbes, untargeted serum metabolomics

## Abstract

Yaks live in the harsh environment of the Qinghai–Tibet Plateau, and the cold climate causes lower growth efficiency. The aim of this experiment was to explore the effects of drinking warm water on the growth performance in yak calves and investigate the underlying physiological mechanisms. A total of 24 Datong yak calves were selected and randomly assigned into the cold water group (group C, water temperature around 0–10 °C without any heating; 58.03 ± 3.111 kg) and the warm water group (group W, water constantly heated at 2 °C; 59.62 ± 2.771 kg). After the 60-day experiment, body weight was measured, and rumen fluid and blood serum samples were collected for analysis. The results show that the body weight and average daily gain of yaks that drank warm water were higher compared to those that drank cold water (*p* < 0.05). The acetic, propionic, isobutyric, valeric, and isovaleric acid concentrations were higher in group W than in group C (*p* < 0.05). Additionally, warm water changed the ruminal microbes at different levels. At the phylum level, the relative abundance of Tenericutes, Kiritimatiellaeota, and Elusimicrobiota was higher in group C (*p* < 0.05). At the genus level, three genera were increased by warm water, including *Ruminococcoides* and *Eubacteriales Family XIII. Incertae Sedis*, and 12 genera were decreased, including *Ruminococcus* (*p* < 0.05). At the species level, unclassified *Prevotellaceae* and *Ruminococcoides bili* were increased by warm water compared to cold water (*p* < 0.05). According to the metabolomics results, metabolites, including valine, isoleucine, PC (15:0/22:2(13Z,16Z)), and LysoPC (18:0/0:0), were increased in the warm water group compared to the cold water group (*p* < 0.05), and were enriched in glycerophospholipid and amino acid metabolism pathways. This study analyzed the differences in ruminal microbes and metabolomes of yak calves provided with water at different temperatures and revealed the potential mechanism for better performance promoted by warm drinking water.

## 1. Introduction

Yaks growing in the harsh environment of the Qinghai–Tibet Plateau in China represent more than 90% of the world’s yak population [1,2]. Due to the particularity of their growth environment and geographical location, there have been few studies on yaks [3,4]. However, with the advancement of science and technology and agricultural modernization, the large-scale breeding of yaks has gradually been included in the regulations. Therefore, the study of the physiology of yaks should be deepened.

The digestive system of yak calves, especially the forestomach, is weaker than that of adult cattle, which causes diarrhea and weight loss [5,6]. According to previous studies, hot water at 35–40 °C can have better effects on hybrid lactating cows in the cold season than warm water at 10–20 °C, and compared with water at 10.18 °C, drinking warm water at 17.69 °C can improve the welfare and performance of beef cattle, significantly increasing average daily gain (ADG) by 30.77% [7,8]. These results indicate that body condition of animals can be improved by increasing the water temperature to some extent. However, whether this method can be adopted for yak calves remains to be investigated.

As a natural fermenter, the rumen plays an important role in feed utilization and energy supply, and there is a biological symbiosis between ruminal microbes and their host; thus, a relatively stable environment in the rumen implies better growth performance of the animal [9]. Rumen microbes can be affected by low environmental temperature [10]. One study showed that the optimal temperature in the rumen was 38.49–38.91 °C, and ruminal microbes in this condition were able to maintain maximal activity, indicating that animals that drink water at a temperature close to the internal environment of the rumen might have preferable outcomes [11]. It was shown that performance was significantly improved with increased ruminal temperature in beef cattle drinking heated water compared with room-temperature water in the cold season [12]. Such results may appear to be similar for yak calves raised in winter.

Ruminal fermentation products such as short-chain fatty acids can enter the bloodstream through the rumen epithelium [13]. Therefore, metabolites can be detected in the serum to assess nutrient utilization in animals [14]. As an omics technology, metabolomics can help to measure the levels of metabolites via a series of biological and mathematical methods [15]. Metabolomics can be used to identify the changes in metabolites caused by different treatments) [16]. Thus, it is useful for evaluating the effects of warm drinking water compared with cold water.

Based on previous findings, we predicted that drinking warm water could improve the performance of yak calves by changing the ruminal microbes and serum metabolites. This study was conducted to determine the effects of warm vs. cold water on yak calves in winter and identify the possible mechanism, so as to provide a theoretical basis for the rapid modernization of yak farming.

## 2. Materials and Methods

### 2.1. Ethics Statement

The study was conducted from 15 January to 15 May 2023, at the Qinghai Yak Breeding and Extension Service Center, Qinghai Province, China. This animal experiment was approved by the China Agricultural University Laboratory Animal Welfare and Animal Experimental Ethical Committee (Permit No. AW22503202-1-1).

### 2.2. Experimental Animals

We selected 24 female Datong yak calves (6 months old) weighing 58.83 ± 2.99 kg and randomly divided the animals into 2 groups: group C: 12 yak calves weighing 58.03 ± 3.11 kg, provided with cold water at around 5 °C without any heating devices; and group W: 12 yak calves weighing 59.62 ± 2.77 kg, provided with warm water at 20 °C.

### 2.3. Experiment Management

The experiment lasted for 2 months in early 2023. The temperature during the experiment was around −9.5 °C, and the relative humidity was 10–90%. The experimental calves were fed twice a day at 9:30 and 16:30 and had access to water ad libitum. Yaks were fed with the total mixed ration (TMR) diet (Table 1).

### 2.4. Sample Collection

Yak calves were weighed on the day before the start of the experiment and again on day 60 to calculate the total weight gain (TWG) and average daily gain (ADG) (using the formula below); all animals were weighed before the morning feeding. On day 61 of the experiment, blood samples were collected from 12 calves (*n* = 6) through the jugular vein using blood collection vessels (Jiangsu Kangjian Medical Apparatus Co., Ltd., Taizhou, China). The blood was centrifuged at 2000× *g* for 10 min. Then, the collected blood serum was stored at −20 °C for later use. On the next day, rumen fluid from 12 calves (*n* = 6) was collected 2 h after the morning feeding, then aspirated through an oral rumen catheter, and filtered through 4 layers of sterile gauze. Later, the fluid was collected into sterile 2 mL cryogenic vials (Wuxi NEST Biotechnology Co., Ltd., Wuxi, China) and stored in liquid nitrogen for later use.
$$ADG=\frac{{Weight}_{day\;60}-{Weight}_{day\;0}}{days\;of\;the\;experiment}$$

### 2.5. Analysis of Ruminal Fermentation Parameters

HPLC-grade N-butanol was placed in a centrifuge tube, and acetic, propionic, butyric, isobutyric, valeric, isovaleric, and caproic acids were added to obtain standard solutions. VFA was analyzed through gas chromatography–mass spectrometry (GC-MS) using Agilent 8890B GC and Agilent 5977B/7000D mass spectrometers. Separation was performed at a flow rate of 1 mL/min an HP-FFAP capillary column (30 m × 0.25 mm × 0.25 µm). The temperature of the gas column was 80 °C and was increased to 120 °C at a rate of 40 °C/min, after which it was increased to 200 °C at a rate of 5 °C/min, and finally held at 220 °C for 3 min. Compounds were identified and quantified using Masshunter software (v10.0.707.0, Agilent, Santa Clara, CA, USA), and the concentrations were calculated.

### 2.6. DNA Extraction and PCR Amplification

Microbial community genomic DNA was extracted from the rumen fluid samples of yak calves using the E.Z.N.A.^®^ Soil DNA Kit (Omega Bio-tek, Norcross, GA, USA), according to the manufacturer’s instructions. The DNA extract was checked on 1% agarose gel, and the DNA concentration and purity were determined with a NanoDrop 2000 UV-vis spectrophotometer (Thermo Scientific, Wilmington, DE, USA). For the bacterial community, the 16S rRNA genes were amplified through the universal bacterial primers 27F (5′-AGRGTTYGATYMTGGCTCAG-3′) and 1492R (5′-RGYTACCTTGTTACGACTT-3′) [17]. The PCR amplification procedure was as follows: initial denaturation at 95 °C for 3 min, followed by 27 cycles of denaturing at 95 °C for 30 s, annealing at 60 °C for 30 s and extension at 72 °C for 45 s, single extension at 72 °C for 10 min, and end at 4 °C (ABI GeneAmp^®^ 9700 PCR Thermal Cycler, Waltham, MA, USA). After electrophoresis, the PCR products were purified using AMPure^®^ PB beads (Pacific Biosciences, Menlo Park, CA, USA) and quantified with a Quantus™ Fluorometer (Promega, Madison, WI, USA). The purified products were mixed in equimolar proportions, and DNA libraries were constructed using SMRTbell^®^ Express Template Prep Kit 2.0 (Pacific Biosciences, USA), according to the manufacturer’s instructions. The purified SMRTbell library was sequenced on the Pacbio Sequel II System (Pacific Biosciences, USA). PacBio raw reads were processed using SMRTLink analysis software (version 8.0) to obtain demultiplexed circular consensus sequence (CCS) reads with a minimum of three full passes and 99% sequence accuracy. CCS reads were barcode-identified and length-filtered. Sequences with a length <1000 or >1800 bp were removed.

The Majorbio Cloud platform (https://cloud.majorbio.com, Accessed on 20 April 2023) was used for bioinformatics analysis of rumen microbes. According to the OTU information, Mothur v1.30.1 [18] was used to calculate the rarefaction curve and α-diversity index, including the Simpson and Shannon indexes. Principal component analysis (PCA) was performed to determine similarities between microorganisms in different groups. The PERMANOVA nonparametric test was used to analyze whether the difference in the microbial community structure between the sample groups was significant. The linear discriminant analysis effect size (LEfSe; http://huttenhower.sph.harvard.edu/LEfSe, Accessed on 20 April 2023) was used to identify the bacterial taxa with significant differences in abundance at the phylum and genus level among groups (LDA score > 2, *p* < 0.05) [19].

### 2.7. Untargeted Metabolome Detection and Analysis

Six biological replicates from two groups were used for LC–MS/MS analysis to evaluate the nontargeted metabolomics, according to a previously described method [20] with minor adjustments.

Metabolites were extracted by adding 400 μL of an acetonitrile:methanol (1:1 (*v*/*v*)) solution containing the internal standard (L-2-chlorophenylalanine) to a 100 μL liquid sample. Vortex mixing and low-temperature ultrasonic treatment were used. The samples were placed at −20 °C for 30 min to precipitate proteins. These samples were then centrifuged for 15 min (4 °C, 13,000× *g*), and the supernatant was removed and blow-dried under nitrogen. The samples were redissolved in an acetonitrile:water (1:1) solution and extracted by low-temperature sonication, and the centrifuged supernatant was subjected to LC-MS analysis. The samples were analyzed by LC-MS/MS on a Thermo Scientific Q Exactive HF-X ultra-performance liquid chromatography (UHPLC)–Fourier transform mass spectrometry system. A 3 μL sample was separated on an HSS T3 column (100 mm × 2.1 mm i.d., 1.8 µm) and entered for mass spectrometry. With a positive ion mode separation gradient of 0–3 min, the mobile phase B increased from 0% to 20%, from 20% to 35% at 3–4.5 min, and from 35% to 100% at 4.5–5 min; then, it was maintained at 100% at 5–6.3 min, decreased from 100% to 0% at 6.3–6.4 min, and maintained at 0% at 6.4–8 min. With a separation gradient of a negative ion mode of 0–1.5 min, the mobile phase B increased from 0 to 5%, from 5% to 10% at 1.5–2 min, from 10% to 30% at 2–4.5 min, and from 30% to 100% at 4.5–5 min; then, it was maintained at 100% at 5–6.3 min, decreased from 100% to 0% at 6.3–6.4 min, and maintained at 0% at 6.4–8 min. The flow rate was 0.40 mL/min, and the column temperature was 40 °C. Positive and negative ion scanning modes were used in a range of 70–1050 m/z. The ion spray voltage in positive and negative mode was set to ±3500 V, and the temperature of the ion transfer tube was 325 °C.

Data processing was performed by Progenesis QI (Waters Corporation, Milford, MA, USA). MS and MSMS information were matched with the public Human Metabolome Database (HMDB; http://www.hmdb.ca/, Accessed on 15 April 2023) and Metlin (https://metlin.scripps.edu/, Accessed on 15 April 2023) to obtain metabolite information. The data matrix was uploaded to the cloud platform (https://cloud.majorbio.com, Accessed on 15 April 2023) for analysis. Principal component analysis (PCA) and orthogonal least partial squares discriminant analysis (OPLS-DA) were performed by R statistical package. The selection of significantly different metabolites was determined based on the variable importance in projection (VIP) obtained by the OPLS-DA model and Student’s *t*-test (VIP > 1, *p* < 0.05) [21]. Differential metabolites of the two treatments were mapped onto their pathways through metabolic enrichment and analysis based on the KEGG database (http://www.genome.jp/kegg/, Accessed on 15 April 2023). Additionally, the correlation of microbes (top 4 with relative abundance) and metabolomes (top 50 with relative abundance) was performed on the cloud platform (https://cloud.majorbio.com, Accessed on 15 April 2023).

### 2.8. Statistical Analysis

GraphPad Prism (version 9.5.0; GraphPad) was used for the unpaired *t*-test analysis. The results are shown as the mean ± standard error; *p* < 0.05 was considered statistically significant, and 0.05 ≤ *p* ≤ 0.1 was considered to be a trend.

## 3. Results

### 3.1. Growth Performance

The weight changes of yak calves over 2 months were measured, and the total weight gain (TWG) and average daily gain (ADG) at end stages were calculated, as shown in Table 2. At the initial stage, two groups of yaks were randomly selected (58.83 ± 2.993 kg), and they showed no significant difference in body weight (BW). The differences in BW between the groups after 60 days (61.32 ± 3.590 and 64.96 ± 3.690 kg for group C and W, respectively) were significant (*p* < 0.05). Moreover, the differences in the TWG (3.283 ± 1.127 and 5.338 ± 1.882 kg for C and W, respectively) and ADG (109.4 ± 37.58 and 177.9 ± 62.74 g for C and W, respectively) were extremely significant (*p* < 0.01).

### 3.2. Rumen Fermentation Parameters

As shown in Table 3, the concentrations of isobutyric, valeric, and isovaleric acid and the A/P of group W were higher than those of group C (*p* < 0.05). In particular, the acetic acid and propionic acid content showed extremely significantly differences between the two groups (*p* < 0.01). In addition, an increasing tendency of NH_3_-N concentration in rumen fluid could be found in the animals drinking warm water (*p =* 0.069).

### 3.3. Ruminal Microbes

A total of 12 samples were collected, including 865,597 optimized sequences and 1,258,226,432 optimized bases, and the average sequence length was 1453 bp. As shown in the rarefaction curves (Figure 1A), the abundance leveled off as the number of reads increased, indicating higher species evenness and richness. A Venn diagram (Figure 1B) shows that the number of OTUs shared by groups C and W was 1686, and each group had 179 and 153 OTUs, respectively, indicating that warm drinking water had an effect on the microbial composition of the rumen fluid. There were no significant differences between the two groups on the ACE and Chao indexes (Table 4), whereas the Shannon and Simpson indexes showed significant differences (*p* < 0.05), which means group W had higher microbiome diversity than group C. PCA at the species level (Figure 1C) showed that there was a significant separation of the confidence ellipses (*p =* 0.033), indicating that the bacteria were different.

The microbial composition at three levels of taxonomic classification was analyzed (Figure 1D–F). At the phylum level, Firmicutes and Bacteroidetes were dominant in both groups. However, the relative abundance of Bacteroidetes in group C was higher, accounting for 46.78% of the total, while Firmicutes accounted for 44.72%. The trend of group W was the opposite, with a relative abundance of 44.96% and 48.84%, respectively. At the genus level, the relative abundance of unclassified order Bacteroides in group C was higher, at 30.74%, followed by the unclassified family Oscillospiraceae, the unclassified order Eubacteriales, and Prevotella. However, the relative abundance of Bacteroides was lower in group W than in group C, accounting for 25.67% of the total. At the species level, the trends of relative abundance of ruminal microbes were similar to those at the genus level. In both groups, the top three microbes with relative abundance were unclassified order Bacteroides, unclassified family Oscillospiraceae, and unclassified order Eubacteriales. However, group W had more *Prevotella ruminicola* (5.11%) than group C (2.89%).

We also analyzed the microbial differences at the same levels (Figure 2A,B and Figure 3A) and found that at the phylum level, microbes from three phyla showed significant differences (*p* < 0.05): Tenericutes, Kiritimatiellaeota, and Elusimicrobia were higher in group C than in group W. At the genus level, 15 genera showed significant differences (*p* < 0.05); the most significant genus was Ruminococcoides, which was more enriched in group W. At the species level, group W had a higher relative abundance of unclassified genus *Prevotell* and *Ruminococcoides bili* than group C. In addition, the linear discriminant analysis effect size (LEfSe) was used to identify the taxa that could explain the differences between the two groups. The significant differences observed in Figure 3B show that yak calves treated with cold water had significantly enriched Rikenellaceae, Ruminococcus, Eubacteriales Family XIII. Incertae Sedis and Tenericutes, while group W had significantly enriched Lachnospiraceae, Ruminococcoides, Butyrivibrio, and Pseudobutyrivibrio.

### 3.4. Untargeted Serum Metabolomics

A total of 662 differential metabolites were identified in the untargeted serum metabolomes in this study, with 421 and 241 metabolites detected in positive and negative ion modes, respectively. As shown in Figure 4A, PCA was performed to evaluate the positive and negative ion data obtained from LC-MS to compare the composition of metabolites of groups C and W. The confidence ellipses did not intersect, and the separation was good. In addition, the OPLS-DA (Figure 4B) showed significant differences between the two groups in positive ions (R2X = 0.268, R2Y = 0.976, Q2 = 0.871) and negative ions (R2X = 0.292, R2Y = 0.973, Q2 = 0.897). The analysis was validated by permutation analysis (positive: Q2 intercept = 0.0141; negative: Q2 intercept = 0.049) (Appendix A). The volcano plot of the positive and negative ion modes is shown in Figure 4C. Based on the cutoff values of differential metabolites (*p* < 0.05, VIP > 1), 204 metabolites were found to be significantly different between groups C and W, among which 119 and 85 were in positive and negative ion mode, respectively (Appendix A).

The heat map in Figure 5A shows great clustering between the two treatments. Moreover, the differential metabolites, mainly composed of phospholipids, fatty acids, and amino acids, were enriched in lipid and amino acid metabolism of metabolic pathways (Figure 5B,C). KEGG topology analysis demonstrated that the top five enriched pathways were cutin, suberine and wax biosynthesis, arginine biosynthesis, glycerophospholipid metabolism, glutathione metabolism and valine, leucine, and isoleucine biosynthesis (Figure 5D).

### 3.5. Correlation of Microbes and Metabolomics

The correlation of differential ruminal microbes (genus level) and serum metabolites was generated as a heat map (Figure 6). An obvious strong correlation between microbes and serum metabolites could be found. The clusters coincided with the above differences between groups C and W, indicating that microbial differences could change the content of serum metabolites. Microbes of *Ruminococcus* and *Eubacteriales Family XIII. Incertae Sedis*, which were more abundant in the rumen fluid of group C, showed negative correlations with PC(15:0/22:2(13Z,16Z)), l-isoleucine, and others, and these metabolites were enriched in lipid and amino acid metabolism. Positive correlations were found between microbes and metabolites, such as 4-hydroxynonenal and L-carnitine, which are implicated in cancer and catabolic processes. However, Ruminococcoides, Butyrivibrio, and Pseudobutyrivibrio, which were more abundant in group W, had the opposite relationship.

## 4. Discussion

This experiment was conducted to investigate the effect of drinking warm water (20 °C) on the growth performance, ruminal fermentation parameters, ruminal microbes, and untargeted serum metabolomes to determine the potential mechanisms and find a more economic and effective way of raising yak calves in the cold season. Because of the weakness of the body in early weaning, all the yak calves were given limited feed, and all the feed was consumed. Thus, there were no effects of feed intake on the results of this experiment.

An animal is an organism with complex functions, and different organs interact with each other to comprehensively affect the growth and development [22]. Regarding the performance of yak calves, the results show that drinking warm water had positive effects on the TWG and ADG. After the 2-month experiment, the calves that drank warm water gained more body weight than those that drank cold water. Additionally, the ADG of calves that drank warm water increased by 62.4% compared with those that drank cold water. According to a previous experiment, the ADG of beef cattle that drank heated water was 1.40 ± 0.39 kg, which was significantly higher than cattle that drank room-temperature water (1.14 ± 0.47 kg) [23]. Another study on beef cattle demonstrated that compared with room-temperature water, drinking heated water could significantly increase the ADG [12]. The results of this study, consistent with those of previous studies, demonstrated that warm water could improve the growth performance of yaks and is available for yak calves in cold periods.

Rumen fermentation parameters, which are used to evaluate the microbial activity and composition, are important indicators of ruminal function [24,25]. Body weight and rumen health would be improved with increased VFAs, particularly acetic and propionic acids [26]. A previous experiment found that the pH of drinking water for sheep at different temperatures showed no differences, consistent with our findings, indicating that generally, changes in water temperature alone cannot alter ruminal pH [27]. In addition, different temperatures in the rumen of dairy cows will cause significant differences in VFAs, and a rumen temperature of 41.33 °C led to the best fermentation compared with 38.03, 39.06, and 43.68 °C [28]. In our study, the concentrations of acetic, propionic, isobutyric, valeric, and isovaleric acids were significantly increased in group W, indicating that the yak calves that drank warm water had better ruminal fermentation.

The composition of the ruminal microbes is essential to the process of ruminal fermentation [29]. This study explored the ruminal microbes and found that there were significant differences at different taxonomic levels. Firmicutes and Bacteroidetes, which mainly participate in the absorption and digestion of carbohydrates and proteins, had the highest abundance at the phylum level, consistent with a previous study [30,31]. Tenericutes, Kiritimatiellaeota, and Elusimicrobia were the three significantly different microbial phyla, which had greater relative abundance in group C. Tenericutes and Elusimicrobia are minor members, and their role in rumen is uncertain [32,33]. According to previous studies, the risk of subacute ruminal acidosis or mycoplasma infection among animals is greater with Tenericutes [34,35], which means the calves in group W were less likely to become sick. Kiritimatiellaeota has been recognized as being involved in fiber digestion [36]. However, a study reported that Kiritimatiellaeota could decrease the dry matter intake (DMI) and metabolic body weight (MBW) of sheep [37]. In this study, the calves in group C did not show greater performance than those in group W, indicating that Kiritimatiellaeota alone cannot increase body weight.

The four most abundant microbes at the genus level were *Ruminococcoides*, *Butyrivibrio*, *Ruminococcus,* and *Eubacteriales Family XIII. Incertae Sedis*. *Ruminococcoides*, as a new genus, is considered to play a key role in degrading resistant starches and providing nutrients and SCAFs for the host [38]. Thus, the higher abundance of *Ruminococcoides* in group W caused better weight gain and rumen fermentation. A previous study demonstrated that members of the genus *Butyrivibrio* were the main species degrading cellulose and hemicellulose, indicating that the calves in group W had a greater ability to digest roughage [39]. *Ruminococcus* has a significant role in the fermentation of polysaccharides [29]. Members of this species, including *Ruminococcus flavefaciens* and *R. albus*, isolated from the rumens of bovines, are cellulolytic, while others, including *R. bicirculans* and *R. callidus*, are non-cellulolytic [40]. Higher *Ruminococcus* levels did not coincide with a lower growth performance in group C, probably because the abundance of *Ruminococcus* was too low (accounting for less than 1% of all ruminal microbes) to make its effect on calves significant. *Eubacteriales Family XIII. Incertae Sedis* showed significant differences between the two treatments. In a study by Passlack et al. [41], the relative abundance of those species was relatively low in the feces of horses given 120 mg/kg zinc methionine. However, there are few studies focused on *Eubacteriales Family XIII. Incertae Sedis*, which needs further investigation.

At the species level, microbes showed differential abundance, including unclassified *Prevotella* and *Ruminococcoides bili*. Members of the genus *Prevotella* have the ability to decompose fiber efficiently, and *Ruminococcoides bili*, which was isolated from a human bile sample, can degrade resistant starches and produce formate, acetate, and lactate [38,42]. The trends of ruminal microbe differences were consistent with the ruminal fermentation characteristics, demonstrating that drinking water heated to a suitable temperature could change the composition of ruminal microbes to some extent and thus increase ruminal function in yak calves.

Drinking water temperature affected the composition of serum metabolites, suggesting that water temperature may alter the metabolites by affecting rumen microbes. The changes in fatty acids, including suberic, caprylic, myristic, and palmitoylcarnitine acids, indicate that drinking warm water could alter the metabolism of fatty acids. According to Valentini [43], fatty acid metabolism is a complex and dynamic process that is affected by multiple biological processes, including synthesis, transport, degradation, and β-oxidation. Medium- and short-chain fatty acids such as octanoic acid directly enter the mitochondria of cells and participate in the tricarboxylic acid (TCA) cycle to provide energy for the body [44]. Long-chain fatty acids require carnitine for transport into mitochondria [45]. In addition, we also observed relatively higher levels of L-carnitine in the serum of calves in group C. Calves that drink cold water in a cold environment need more energy to resist stress than those that drink warm water, and higher levels of L-carnitine could promote fatty acid degradation, which would explain the lower body weight of yaks in group C and the better growth of yaks in group W. Additionally, a study by Liu et al. [22] reported greater apparent total tract digestibility of ether extract in animals that drank heated water. Thus, we suppose that calves provided with warm drinking water were better able to utilize fatty acids.

The content of most glycerophospholipids in the serum of calves in group W was relatively higher compared to group C, which means the level of glycerophospholipid metabolism was higher. However, it is unclear that whether drinking warm water could promote the synthesis or degradation of glycerophospholipids based on the results of the untargeted metabolome in this study. PC(18:0/22:4(7Z,10Z,13Z,16Z)), PC(15:0/18:3(9Z,12Z,15Z)), PC(22:5(7Z,10Z,13Z,16Z,19Z)/18:0), PC(15:0/22:2(13Z,16Z)), and PE(20:4(8Z,11Z,14Z,17Z)/18:0) were significantly upregulated in group W. Studies have shown that phosphatidylcholine and 2-lysophosphatidylcholine can be converted into each other by lysophosphatidylcholine acyltransferase or lecithin cholesterol acyltransferase catalysis, and choline, the precursor of acetylcholine, can be generated [46,47]. Phosphatidylcholine is not only an important component of biofilms but also a key molecule in the transmission of information, regulation of apoptosis, promotion of fat metabolism, and elimination of cholesterol in serum. Additionally, both phosphatidylcholine and phosphatidylethanolamine are antioxidants, suggesting that yak calves that drink warm water have better antioxidant capacity [46,48]. As a result, we propose that these metabolites could be used as potential targets for the regulation of phospholipid metabolism in yak calves.

In this study, significant differences were found in the concentrations of amino-acid-related metabolites in the serum of calves that drank water at different temperatures. As the precursors for protein synthesis, amino acids are mainly produced by the microbial degradation of protein in food, and they play an important role in the metabolic regulation process [49,50]. In group W, upregulation of valine, isoleucine, and ornithine acid and downregulation of glycine were observed. It has been shown that glycine can be fermented to acetate by the glycine reductase system in microbes [51]. Thus, more glycine is degraded in the rumen, resulting in higher amounts of acetate in the rumen and lower amounts of glycine in the blood. The higher acetate concentration in group W was consistent with this rumen fermentation characteristic, indicating that the result is credible. In addition, both L-leucine and L-valine in blood are preferentially involved in the TCA cycle [52]. Therefore, yaks in group W might have had more energy for growth. A previous study confirmed that ammonia is produced in the rumen and bound to aminomethyl phosphate in the liver, and later reacts with ornithine to form citrulline, and a higher ornithine acid content is consistent with higher levels of ammonium-N detected in the rumen [53]. Therefore, these results suggest that the digestive system of calves in the warm water group was more efficient at using protein from their diets.

The top four genera and top 50 serum metabolites were associated, and a heat map of the correlation was generated. It is not difficult to see a strong correlation between the ruminal microbes and serum metabolites. Ruminococcoides and Butyrivibrio showed positive correlations with metabolites such as PC(15:0/22:2(13Z,16Z)), LysoPC(18:0/0:0), and PE-NMe2(18:1(11Z)/18:0) and negative correlations with butyryl L-carnitine, L-carnitine, hippuric acid, benzoic acid, and various fatty acids. However, *Eubacteriales Family XIII. Incertae Sedis* and *Ruminococcus* had the exact opposite trend. Studies have shown that ruminal microbes can alter the phenotypic traits of ruminants, and ruminal temperature can influence the microbes, in turn altering the metabolites [11,54]. These changes indicate the interaction between the two treatments (cold and warm drinking water) and ruminal microbes and serum metabolites, leading to changes in the growth performance of yak calves. However, the combined effects are complex, and the way in which drinking temperature affects ruminal microbes and the potential mechanisms need further investigation.

## 5. Conclusions

This study found that providing warm drinking water (20 °C) for yak calves in winter could promote ruminal fermentation, increase the relative abundance of microbes such as Ruminococcoides and Butyrivibrio, promote fatty acid and amino acid synthesis, and ultimately improve the growth performance.

## Figures and Tables

**Figure 1 microorganisms-11-02092-f001:**
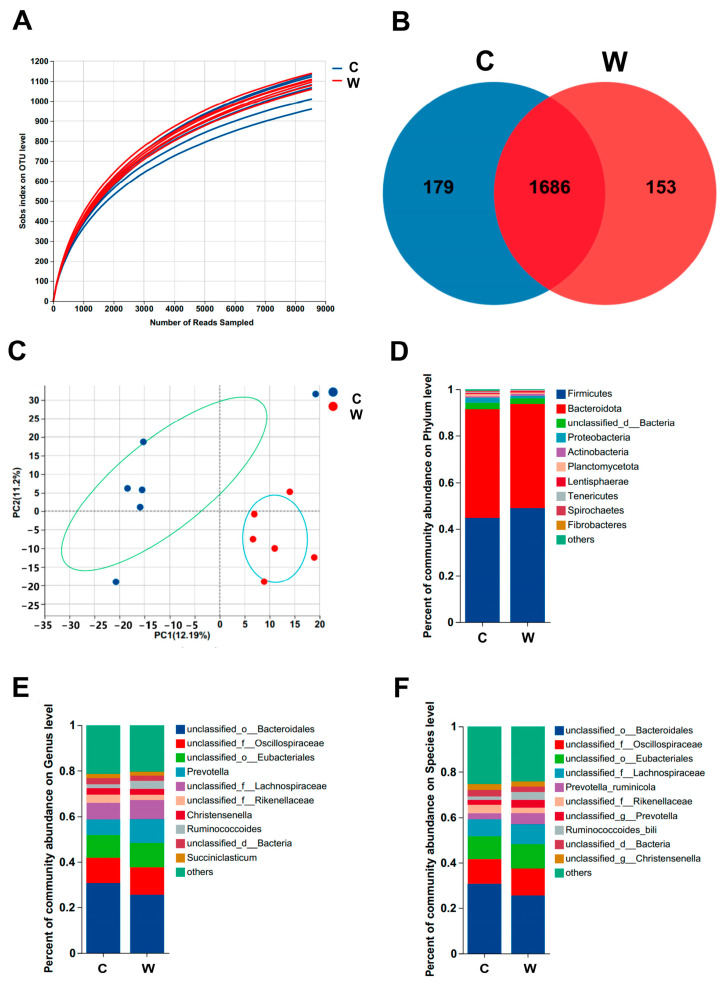
Effects of warm water on the composition of ruminal microbes (*n* = 6). (**A**) Rarefaction curves of rumen samples. (**B**) Venn diagram of OTU level in two treatments. (**C**) Principal component analysis (PCA) plots of rumen samples at species level (R^2^ = 0.1796, *p* = 0.033). (**D**–**F**) Stacked bar graphs of the average relative abundance in two treatments at phylum, genus, and species level, respectively.

**Figure 2 microorganisms-11-02092-f002:**
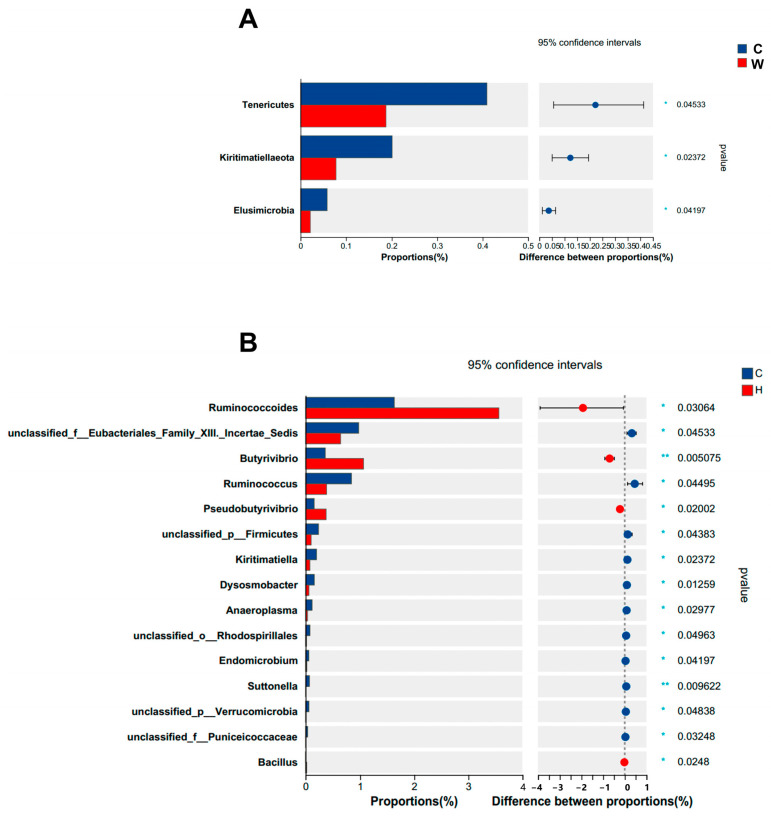
Effects of warm water on the differences in the ruminal microbes (*n* = 6). Difference map of the relative abundance of the two treatments at (**A**) phylum level and (**B**) genus level.

**Figure 3 microorganisms-11-02092-f003:**
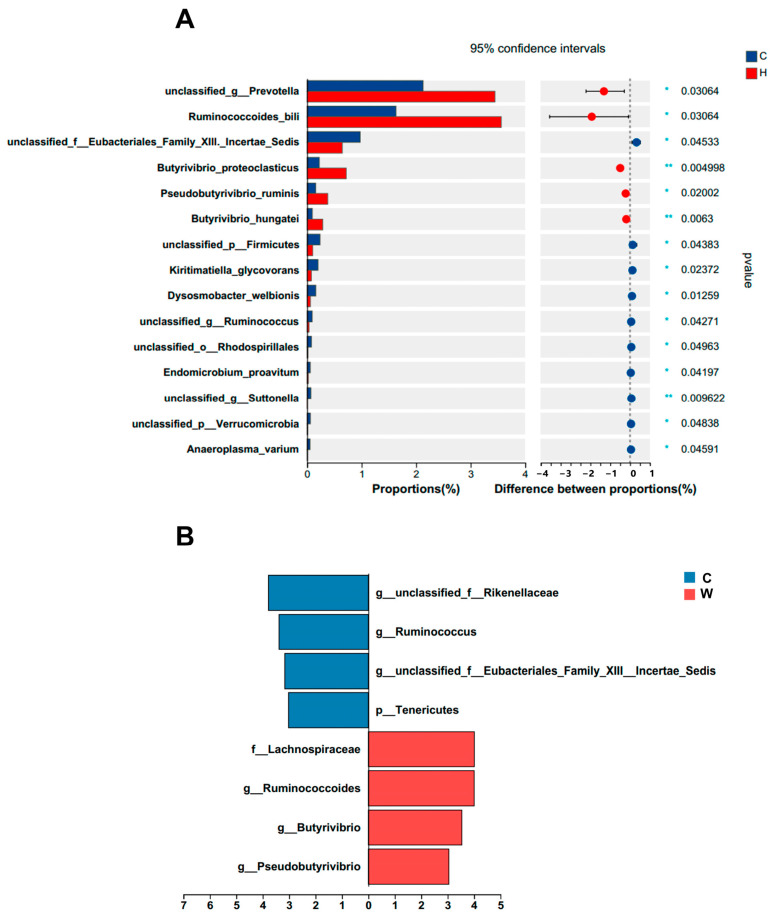
Effects of warm water on the differences in the ruminal microbes (*n* = 6). (**A**) Difference map of the relative abundance with the two treatments at species level. (**B**) LEfSe analysis of two treatments at phylum to genus level (LDA ≥ 3).

**Figure 4 microorganisms-11-02092-f004:**
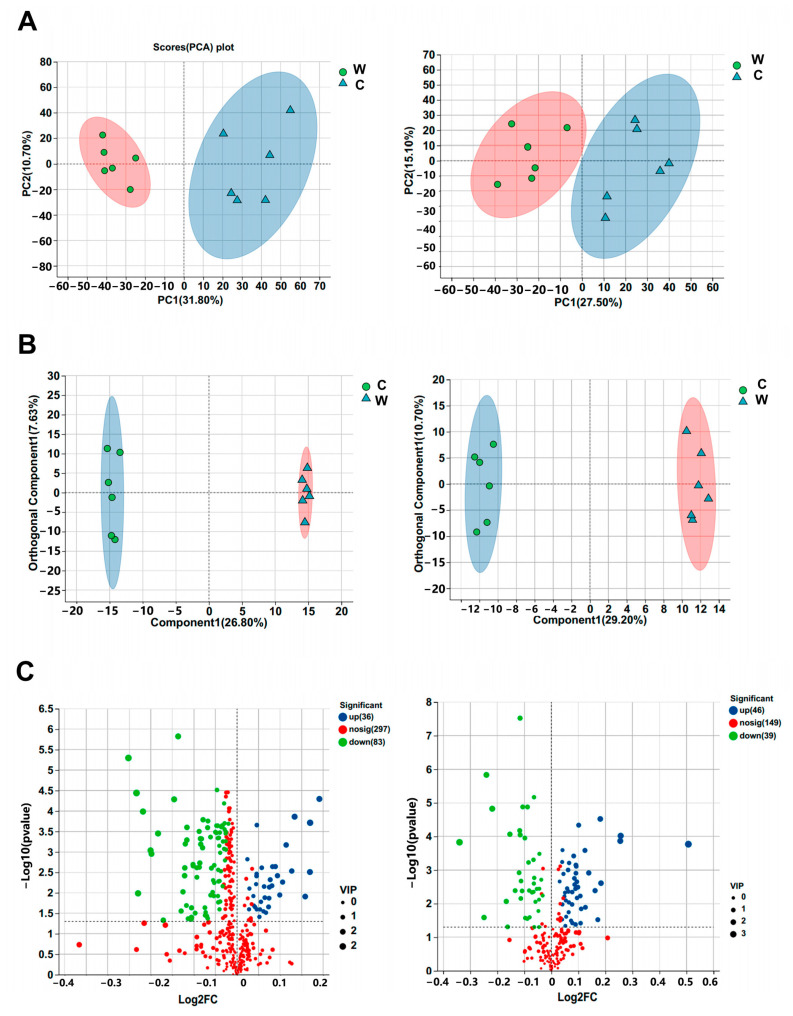
Metabolomic profiles of serum samples (*n* = 6). (**A**) PCA of positive and negative ion modes; (**B**) OPLS-DA of positive and negative ion modes; (**C**) volcano plots of different metabolites of positive and negative ion modes.

**Figure 5 microorganisms-11-02092-f005:**
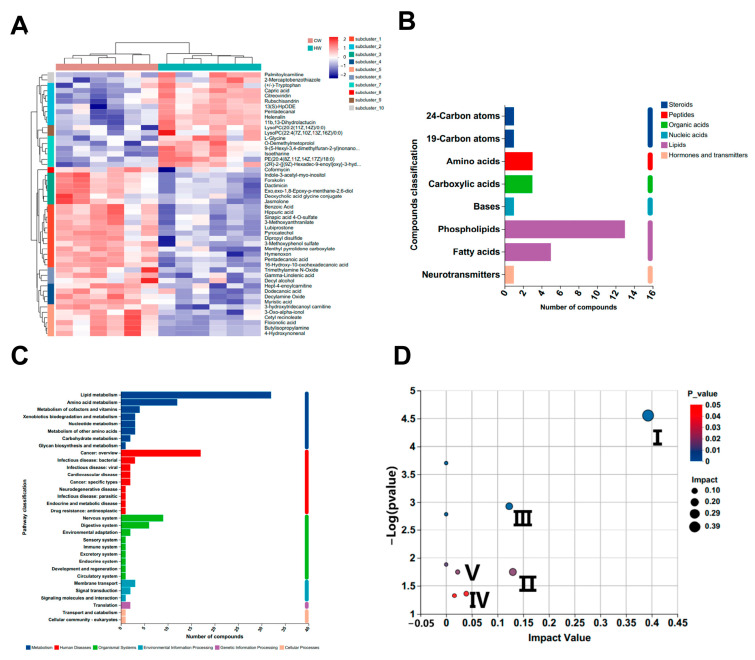
Metabolomic profiles of serum samples (*n* = 6). (**A**) Hierarchical clustering analysis of metabolites. (**B**) Metabolite compound classification. (**C**) Enriched KEGG pathways of metabolites. (**D**) KEGG topology analysis. I: cutin, suberine, and wax biosynthesis; II: arginine biosynthesis; III: glycerophospholipid metabolism; IV: glutathione metabolism; V: valine, leucine, and isoleucine biosynthesis.

**Figure 6 microorganisms-11-02092-f006:**
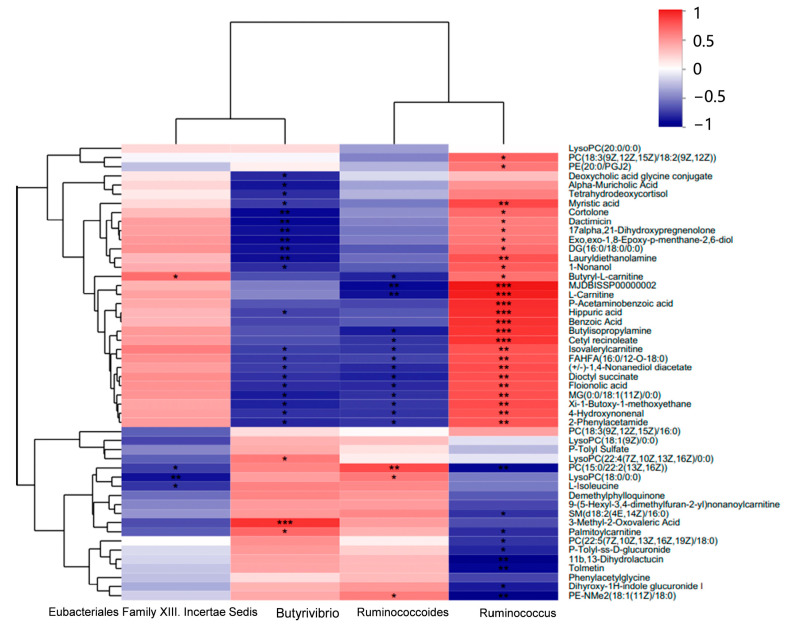
Correlation analysis of the ruminal microbes and metabolites (*n* = 6). The names of metabolites are shown on the right side; the names of microbes are on the bottom; different colors represent the correlation coefficient size between attributes (red for positive and blue for negative correlations). *** Extremely significant correlation (*p* < 0.001); ** extremely significant correlation (*p* < 0.01); * significant correlation (*p* < 0.05); no asterisk indicates nonsignificant correlation.

**Table 1 microorganisms-11-02092-t001:** Ingredients and nutrients of the experimental diet.

Item	Amount
Ingredients (%)	
Dried wheat grass	37.0
Dried oat grass	32.0
Corn silage	17.0
Wheat	4.0
Wheat bran	2.5
Rapeseed meal	4.0
Soybean meal	1.5
NaCl	0.5
Premix ^1^	1.5
Chemical composition (% of dry matter (DM))	
DM	86.32
Crude protein (CP)	6.25
Ether extract (EE)	1.28
Neutral detergent fiber (NDF)	54.26
Acid detergent fiber (ADF)	31.57
Coarse ash (CA)	9.50

Notes: ^1^ Premix provides the following per kilogram of feed: Ca 6 g, P 4.5 g, Na 4.8 g, K 9 g, Mg 3.6 g, Cu 22 mg, Fe 126.3 mg, Zn 46.8 mg, Mn 54.18 mg, vit A 60,000 IU, vit D3 25,000 IU, vit E 100 IU. Nutrient levels were measured values.

**Table 2 microorganisms-11-02092-t002:** Effects of warm water on growth performance of yak calves (*n* = 12).

Items	Group C	Group W	*p*-Value
Body weight _day 0_ (kg)	58.03 ± 3.11	59.62 ± 2.77	0.200
Body weight _day 60_ (kg)	61.32 ± 3.59	64.96 ± 3.69	0.023
Total weight gain (kg)	3.28 ± 1.13	5.34 ± 1.88	0.004
Average daily gain (g)	109.40 ± 37.58	177.90 ± 62.74	0.004

**Table 3 microorganisms-11-02092-t003:** Effects of warm water on the ruminal fermentation characteristics in yak calves (*n* = 6).

Items	Group C	Group W	*p*-Value
pH	6.97 ± 0.10	6.86 ± 0.28	0.402
NH_3_-N (mg/kg)	68.33 ± 71.95	186.70 ± 122.40	0.069
VFA (mg/kg)			
Acetic acid (A)	2373.00 ± 189.50	2849.00 ± 241.90	0.004
Propionic acid (P)	848.20 ± 93.66	1141.00 ± 112.80	0.009
Butyric acid	681.00 ± 73.85	766.80 ± 96.80	0.115
Isobutyric acid	41.47 ± 13.19	59.95 ± 6.06	0.011
Valeric acid	64.05 ± 17.90	86.45 ± 13.58	0.035
Isovaleric acid	57.75 ± 21.53	79.23 ± 8.19	0.045
Hexanoic acid	20.32 ± 9.06	16.85 ± 7.99	0.498
A/P	2.81 ± 0.25	2.50 ± 0.13	0.020

**Table 4 microorganisms-11-02092-t004:** Effects of warm water on the characteristics of yak calves (*n* = 6).

Index	Group C	Group W	*p*-Value
ACE	1377 ± 84.77	1416 ± 29.38	0.318
Chao	1373 ± 82.97	1417 ± 33.59	0.256
Shannon	5.818 ± 0.1278	5.962 ± 0.0857	0.045
Simpson	0.008914 ± 0.002306	0.006136 ± 0.0007554	0.019

## Data Availability

The data presented in this study are available on request from the corresponding author.

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
