# Peer review of "Drinking Warm Water Promotes Performance by Regulating Ruminal Microbial Composition and Serum Metabolites in Yak Calves"

_microorganisms, 2023, doi:10.3390/microorganisms11082092_

Round 1

Reviewer 1 Report

The study provides interesting findings about the growth performance and ruminal microbiota in yaks, as affected by the warm water drinking. The studies on yaks remain scarce, however, with the need to modernize the agriculture, the breeding of yaks has been expanding and hence the need of research on this species becomes more evident. The authors have clearly defined the aim of their study. The Introduction provides sufficient background to convince in the need of this particular study. The research design is appropriate, however, there are some issues, I would like to remark. 1. Why have the authors opted to do the experiment with female animals?

2. It would be good to show the diet composition and nutritional value.

3. Why the blood samples and ruminal samples were taken from only 6 animals? This is minimum for statistics.

4. Are the animals subjected to blood and ruminal sampling the same?

5. Please describe that you have dome PCA and correlation in the section concerning the statistical evaluation. Also, R is not a language, it is a statistical package.

The results are clearly described and well discussed. The conclusions, although very concise are sound and derived from the results.

Reviewer 2 Report

The manuscript need a thorough editing.  I will make a few suggestions for its improvement.

line 51:  Change increasement to increase here and elsewhere.

line 53-56:  Rewrite for clarity.

lines 65-69:  Rewrite to improve clarity.  Change serious to series.

line 72:  Delete in a widely range at end of sentence.

line 87:  Rewrite to not use 24 as opening of sentence.

line 88:   Change weighted to weighing here and in next line.  What was the average temperature of the warm and cold waters?  Decrease number of significant figures in weight data.

line 95:  Please describe the TMR.  What was its composition?

lines 97 o 107:  Rewrite to improve clarity.  Change rpm to x g.

lines 120 to 124:  Rewrite for improved clarity.

lines 135 and 136:  Rewrite for clarity.  Were data really filtered?

line 137:  A sequence of 180 bp is not a sentence.

line 140:  Delete the vague and so on.

line 143-146:  Rewrite for clarity.

lines 148 to 181:  A reference to this procedure is needed.  Or,  is the procedure original with authors?   Rewrite this section to condense and remove unnecessary details for a referenced procedure.

Table 2:  Units of NH3 and VFA are unclear.  To what does kg refer?

line 216:  Insert that of after than.

Figures 1 through 6:  Figures are 'unreadable' because of small size of printing.  Can that be corrected?

line 286 to 292:   Reconsider the use of metabolism here and throughout manuscript.  Remember that metabolism includes both synthesis and degradation (oxidation). 

lines 300 to 306:  Rewrite for improved clarity.  

line 308:  Do not capitalize names of compounds when within the sentence here and throughout manuscript.  Delete the vague and so on in this sentence.  What is PC in line 307?

lines 306 to 309:  Rewrite for clarity.  What do you mean by enriched in lipid and amino acid metabolism?

lines 323-324:  Rewrite for improved clarity.

line 337 to 339:   Rewrite for improved clarity.  What is upregulation of water temperature?

line 361:  What do you mean by  up to date?

line 365:   Be more precise than better fitness.

line 375:  Be more specific than better performance.

line 389:  Investigate to investigation.

line 391:  Delete the vague etc.

line 412:  Do you mean glycerophospholipids in serum?

lines 420 to 425:  Rewrite for improved clarity.

Discussion needs to include impact of warm water on feed intake.  All improvements caused by warm water may have been the result of increased feed intake.

lines 433-435:  Rewrite for improved clarity.

line 461:  Rewrite for improved clarity.  What is promoted glycerophospholipids and amino acid metabolism?

line 438:  What is better energy metabolism?

Manuscript requires extensive rewriting.  I offered a few suggestions.

Round 2

Reviewer 2 Report

I consider the modified manuscript ready to publish.

Thanks for markedly improving the manuscript.